# Diagnostic Evaluation of an Increased Risk of Developing Small Intestinal Bacterial Overgrowth Associated with Glucagon-like Peptide-1 (GLP-1) Receptor Agonists and Dual GLP-1/GIP Receptor Agonists: A Global Retrospective Multicenter Cohort Analysis

**DOI:** 10.3390/diagnostics15172264

**Published:** 2025-09-07

**Authors:** Yan Sun, Donovan Veccia, Benjamin Douglas Xun Liu, William Tse, Ronnie Fass, Gengqing Song

**Affiliations:** 1Department of Medicine, MetroHealth Medical Center, Case Western Reserve University, Cleveland, OH 44109, USA; ysun1@metrohealth.org (Y.S.); bliu257@uwo.ca (B.D.X.L.); 2Division of Gastroenterology and Hepatology, MetroHealth Medical Center, Case Western Reserve University, Cleveland, OH 44109, USA; dveccia@metrohealth.org (D.V.); rfass@metrohealth.org (R.F.); 3Division of Hematology and Oncology, MetroHealth Medical Center, Case Western Reserve University, Cleveland, OH 44109, USA; wtse@metrohealth.org

**Keywords:** GLP-1 receptor agonists, small intestinal bacterial overgrowth, diagnostic evaluation, breath test, SIBO, diabetes mellitus, gastrointestinal motility, dual GLP-1/GIP receptor agonists, Trinetx database

## Abstract

**Background/Objectives:** Glucagon-like peptide-1 receptor agonists (GLP-1 RAs) effectively manage type 2 diabetes mellitus (T2DM) but may impair gastrointestinal motility, increasing the risk of small intestinal bacterial overgrowth (SIBO). Diagnostic evaluation of SIBO commonly involves breath testing and clinical assessment. This study aimed to assess the association between GLP-1 RAs or dual GLP-1/glucose-dependent insulinotropic polypeptide (GIP) RAs are associated with incident SIBO. **Methods:** We conducted a retrospective cohort study using the TriNetX global database, identifying adult T2DM patients initiating GLP-1 RA or dual GLP-1/GIP RA therapy versus other second-line T2DM agents (OSLT2DM) from 1 January 2006 to 2 December 2024. Patients with major abdominal surgery, connective tissue disorders, gastroparesis, or other high-risk conditions for SIBO were excluded. 1:1 Propensity score matching was applied. Short-term (<1 year) and long-term (up to 5 years) risks were evaluated with Kaplan–Meier curves and univariable Cox models. **Results:** After matching, 216,173 patients per cohort were analyzed. Short-term analysis demonstrated a higher incidence of diagnostically confirmed SIBO in patients treated with GLP-1 RA/GIP (0.177 per 1000 patient-years) compared to OSLT2DM (0.083 per 1000 patient-years; HR 2.14, 95% CI 1.13–4.07; *p* = 0.0491). Long-term analysis indicated a non-significant trend toward increased risk in the GLP-1 RA/GIP group (HR 2.02, 95% CI 0.98–4.12), though Kaplan–Meier analysis revealed a sustained divergence (*p* = 0.017). **Conclusions:** GLP-1 RA and dual GLP-1/GIP RA therapy are associated with increased short-term SIBO risk. Symptom-driven SIBO breath-test evaluation may be warranted in patients initiating these agents.

## 1. Introduction

Glucagon-like peptide-1 (GLP-1) and glucose-dependent insulinotropic polypeptide (GIP) are incretin hormones naturally produced in the body that help regulate blood glucose levels after eating [1]. FDA-approved GLP-1 receptor agonists (GLP-1 RAs), including liraglutide, lixisenatide, exenatide, dulaglutide, and semaglutide, are widely prescribed in the United States for the treatment of type 2 diabetes mellitus (T2DM) and complications [2,3]. A newer dual glucose-dependent insulinotropic peptide (GIP) and GLP-1 receptor agonist, tirzepatide, is also approved for treating type 2 diabetes mellitus [2,4]. Their major benefits include effective glycemic control, weight loss, lowering of blood pressure and cholesterol, and improvement of renal function [5].

Despite their effectiveness, GLP-1 RAs have several negative gastrointestinal side effects, including nausea, vomiting, diarrhea, pancreatitis, bowel obstruction, and notably gastroparesis, which is associated with small intestinal bacterial overgrowth (SIBO) [6,7]. Recently, there was one case report that found semiglutide caused patient lactulose and gluten intolerance. 6 months after semiglutide discontinuation, the patient was diagnosed with SIBO [8]. Although not fatal by itself, SIBO can lead to serious complications as weight loss, malnutrition, absorption, diarrhea, bloating, and nausea, which significantly impact patients’ daily lives [9,10,11].

The primary goal of this study was to diagnostically assess the connection between GLP-1 RA usage and the incidence of SIBO, quantified through rates of clinically confirmed cases. Secondary objectives included determining the frequency of hydrogen breath tests used for diagnosis, the average duration of follow-up after diagnosis, and the proportion of patients initiating targeted treatment within one week following SIBO diagnosis.

## 2. Materials and Methods

### 2.1. Data Source

We conducted a retrospective cohort study using the TriNetX platform (TriNetX, 100 Cambridgepark Drive, Suite 501, Cambridge, MA, 02140, USA), a de-identified multicenter, global electronic health record (EHR) database [12]. TriNetX includes data on >300 million patients from participating healthcare organizations across 19 countries, and supports de-identified, line-level analyses within the platform, while ensuring complete patient anonymity. To protect privacy, occurrences involving 10 or fewer patients are uniformly suppressed and reported as “≤10”, and exact counts below this threshold are not disclosed. [12]. The MetroHealth Medical Center Institutional Review Board (IRB) determined this study to be exempt from IRB approval because it uses only de-identified and aggregated data, complying with HIPAA Privacy Rule (§164.514(a)). This study adheres to the Strengthening the Reporting of Observational Studies in Epidemiology (STROBE) guidelines for cohort studies; a completed checklist is provided in the Appendix A (Appendix A) [13].

### 2.2. Patient Selection

We identified adults (>18 years) with type 2 diabetes mellitus (T2DM) in the TriNetX database (Figure 1). The index event was the first observed prescription on or after 1 January 2006. Because the first GLP-1 receptor agonist (exenatide) received U.S. FDA approval in 2005, we began cohort accrual on 1 January 2006 to ensure capture of initiators in the post-approval period [6,14]. The prescription was either (a) a GLP -1 RAs (Exenatide, liraglutide, lixisenatide, dulaglutide, semaglutide) or the dual GLP-1/GIP RA tirzepatide (GLP RA/GIP cohort) or (b) another second-line antihyperglycemic agent (OSLT2DM cohort: sodium-glucose cotransporter-2 inhibitors (SGLT2), thiazolidinediones (TZD), dipeptidyl peptidase 4 inhibitors (DPP-4), and sulfonylureas, chosen for their prevalent use in similar populations without known associations with SIBO. Cohorts were mutually exclusive and assigned by the index agent. Query search criteria were based on standard diagnostic, procedure, or medication codes.

Exclusion criteria included patients younger than 18 years, any patient with a death date on or before the index date (deceased patients at baseline), individuals with pre-existing conditions known to impact gastrointestinal GI functions, e.g., major GI or abdominal surgeries, connective tissue disorders (e.g., amyloidosis, Ehlers-Danlos syndrome, Sjögren’s syndrome), inflammatory bowel diseases, and other major systemic or GI conditions (cystic fibrosis, intestinal failure, etc.) predisposing to SIBO. Patients with pre-existing gastroparesis prior to medication initiation were also excluded. Code lists are provided in Table 1.

Cohort definitions: We identified adult T2DM initiators of GLP-1 RA or dual GLP-1/GIP RA and OSLT2DM agents; cohort assignment was fixed by the index agent and not redefined thereafter. The same initiator cohorts were used for all analyses; “short-term” refers to evaluations at 3, 6, and 12 months, and “long-term” refers to evaluations after 3 months of initiation and observed for 60 months. Outcome definitions and censoring rules are described in Section 2.5. In the long-term cohort, we continuously monitored patients from initiation of prescription to 60 months.

### 2.3. Query Diagnostic Validation

We validated our database queries through an unmatched analysis of patients exposed to GLP-1 RAs, dual GLP-1/GIP RA, and OSLT2DM. We identified patients who received a new SIBO diagnosis after initiation and, within this SIBO subset, recorded the proportion who had a hydrogen breath test documented that those who received SIBO-direct antibiotics therapy within 7 days of the first SIBO code. Breath tests have moderate sensitivity and protocol-dependent performance; substrate and gas values were not uniformly available in TriNetX. Any resulting misclassification is likely non-differential between matched cohorts and would attenuate observed associations.

### 2.4. Covariates and Matching

We identified potential confounders related to SIBO risk and the selection of T2DM therapy, including age at index, sex, BMI, HbA1C, hypertension, hyperlipidemia, atherosclerotic cardiovascular disease, hypothyroidism, medication such as opioids, proton pump inhibitors (PPI), etc. (Table 2). We estimated the propensity to initiate GLP-1/GLP-GIP RAs versus OSLT2DM (Table 2). Dietary or probiotics intake data were not available in TriNetX; thus, diet could not be included as a covariate.

Continuous variables were summarized as mean (SD) and compared pre-match using Welchi’s *t*-test; categorical variables were summarized as n (n%) and compared using Pearson’s X (or Fisher’s exact when expected counts were <5). After matching, covariate balance was assessed using absolute standardized mean differences (SMD), with SMD < 0.10 indicating adequate balance. For cells with ≤observations, *p*-values were suppressed per data-use policy; exact tests were used for interference where applicable (Appendix A).

### 2.5. Propensity-Matched Study Outcomes

The outcome was diagnostic confirmation of SIBO: Using the propensity-matched initiator cohorts, we assessed incident SIBO beginning the day after the index prescription for 3, 6, or 12 months (short term). We also conducted a landmark analysis at 3 months: patients who were event-free and had continuous exposure ≥3 months were included; time zero was reset to the 3-month landmark, and outcomes were assessed from 3 to 60 months using KM and univariable Cox models.

### 2.6. Statistical Analysis

Statistical analyses were performed using the TriNetX platform or R software (version 4.4.2; RStudio). Descriptive statistics included means, standard deviations (SD), and proportions. Propensity score matching was conducted 1:1 using relevant covariates through greedy nearest neighbor algorithms, with a caliper width of 0.1, ensuring well-matched cohorts [15]. Characteristics with a standard mean difference between cohorts lower than 0.1 were described as well-matched [15]. Time-to-event analyses on the matched cohort used Kaplan–Meier estimates with log-rank tests [16]. Patients were observed until the diagnostic confirmation of SIBO, last clinical encounters, death, or the analysis horizon, as described above, whichever occurred first. Hazard ratios (HRs) were calculated using univariate Cox-proportional hazards models on the TriNetX platform, with statistical significance defined by a *p*-value of less than 0.05. Incidence rates were calculated per 1000 person-years; between-group comparisons of incidence used Poisson regression (log link) with an offset for log (person-time), reporting incidence rate ratios (IRR) with 95% CI and Wald *p*-value. Figures were generated in Microsoft Excel (Redmond, WA, USA, version 2410), R 4.4.2 (Vienna, Austria), and RStudio (version 2024.09.1-394, Boston, MA, USA).

## 3. Results

### 3.1. Study Population

From the source population of 163,617,518 patients [12], we identified 9,159,059 adult patients diagnosed with T2DM. Among these, 1,281,479 patients initiated therapy with either GLP-1 RA, dual GLP/GIP RA (GLP-1 RA/GIP), or OSLT2DM between 1 January 2006 and 2 December 2024. After applying exclusion criteria, the GLP-1 RA or dual GLP/GIP RAs cohort comprised 256,362 patients, while the control cohort consisted of 1,025,117 patients (Figure 1). After performing one-to-one propensity score matching, each matched cohort included 216,173 patients. Demographic features and comorbidities of the matched cohorts are detailed in Table 2. All baseline characteristics were well-balanced post-matching [15]. Figure 1 summarizes cohort selection for patients with extended GLP-1 RA/GIP or OSLT2DM treatment durations.

### 3.2. SIBO Incidence Rate in Short-Term and Long-Term Outcome

SIBO incidence is reported per 1000 person-years (PY); between-group differences are summarized as incidence rate ratios (IRR) with 95% CIs and two-sided *p*-values. For 3 months, GLP-1/dual GLP-1–GIP RAs: incidence 0.791/1000 PY vs. OSLT2DM: incidence 0.575/1000 PY; IRR 1.38 (95% CI 0.72–2.62), *p* = 0.332. For 6 months, GLP-1/dual GLP-1–GIP RAs: incidence 0.327/1000 PY vs. OSLT2DM: incidence 0.251/1000 PY; IRR 1.30 (95% CI 0.57–2.96), *p* = 0.533. For 12 months, GLP-1/dual GLP-1–GIP RAs: incidence 0.177/1000 PY vs. OSLT2DM: incidence 0.083/1000 PY; IRR 2.90 (95% CI 1.41–5.95), *p* = 0.0037. Followed 3 up to 60 months, GLP-1/dual GLP-1–GIP RAs: incidence 0.0758/1000 PY vs. OSLT2DM: incidence 0.0339/1000 PY; IRR 1.46 (95% CI 0.72–2.96), *p* = 0.292. In summary, in these persistence-restricted analyses, estimates at 3 and 6 months were attenuated and not statistically significant, whereas the 12-month subset showed a significant association.

### 3.3. Kaplan–Meier Analyses: Univariable Cox Proportional Hazards

Kaplan–Meier curves and univariable Cox models showed: 3 months: HR 1.786, 95% CI 0.942–3.386, log rank *p* = 0.072, which means higher hazard in GLP-1 RA/GIP, but not statistically significant. 6 months: HR 1.867, 95% CI 0.973–3.585, log rank *p* = 0.054, means borderline separation still not significant. 12 months: HR 2.140, 95% CI 1.13–4.07, log rank *p* = 0.049, the difference is significant. In the 3-month landmark analysis, Kaplan–Meier analyses demonstrated a statistically significant divergence beginning approximately 90 days post-treatment initiation and continuing over 60 months (822.44 ± 539.54 days, compared to 1256.15 ± 593.14 days, median days were 715 vs. 1514): HR 2.018, 95% CI 0.989–4.116, log rank *p* = 0.0173. Although the log rank indicated a significant between-group difference over 3–60 months, the Cox hazard ratio showed a non-significant (borderline) increased SIBO risk in the GLP-1 group (Figure 2).

## 4. Discussion

### 4.1. Epidemiology and Therapeutic Context

Approximately 30.3 million Americans have diabetes, 90–95% of whom have T2DM [17]. Since 2005 the first GLP-1RA was approved, around 679,265 individuals in the U.S. were treated with GLP-1 RAs as of 2023 [18]. Tirzepatide, the first dual GLP/GIP receptor agonist, was approved in 2022 [19]. Beyond glycemic control in T2DM, GLP-1 RA and tirzepatide are increasingly used in obesity [20,21]. Obstructive sleep apnea (OSA) [22], heart failure [23]. Concomitantly, multiple studies have reported gastrointestinal (GI) adverse events with GLP-1 RAs-including nausea, diarrhea, vomiting, constipation, abdaominal pain, and pancreatitis [24]. Recently, Amrutha S et al. used Facebook to investigate the adverse effects of GLP-1 RAs and tirzepatide. The results have also highlighted GI complaints with GLP-1 RAs. The GI symptoms, such as nausea, vomiting, pancreatitis, and diarrhea, were strongly associated together [25].

### 4.2. SIBO Background and Diagnostic Approch

SIBO (small intestinal bacterial overgrowth) is characterized by bacterial overgrowth or colonization of the small intestine, accompanied by GI symptoms such as bloating, diarrhea, nausea, malabsorption, and food intolerance [11]. Diagnostic options include a hydrogen breath test (non-invasive) and quantitative small bowel aspiration culture (invasive), with >103 colony-forming units (CFU)/mL commonly used as a threshold in duodenal aspiration. In this study, we used a hydrogen breath test to diagnose SIBO because it is practical and non-invasive [26]. Given the limitation of aspirating culture, a breath test is the most practical alternative [27].

### 4.3. Study Outcomes: Interpretation and Limitations

Direct evidence linking GLP-1 RAs and GLP-1/GIP RA to SIBO is limited; we therefore examined this association in a large, propensity-matched cohort. In the short-term windows, the hazard was higher at 3 and 6 months but did not reach statistical significance (few early events → wide CIs). By 12 months, the curves separated, and the association was significant, indicating a sustained difference within the first year. In contrast, the 3–60-month landmark window showed an attenuated/borderline association: excluding the early post-initiation period—when the difference is greatest—reduces the average contrast over the longer window. These patterns are expected for three reasons: (i) early windows have low event counts, yielding wide CIs and larger *p*-values; (ii) conditioning on persistence (patients who remain on therapy) preferentially keeps individuals who tolerated treatment and stayed event-free early (“depletion of susceptible”), which shrinks between-group differences; and (iii) starting follow-up at 3 months omits the initial high-risk period, so long-window averages are diluted. Taken together, the results are consistent with an early divergence that strengthens by 12 months but becomes less pronounced when early time is excluded.

### 4.4. Mechanisms Hypothesis

The mechanisms linking GLP-1 RA, GLP/GIP RA, and SIBO remain uncertain. Two hypotheses are most reasonable: delayed GI transit, increased bacterial proliferation. GLP-1 RA can reduce gut motility by slowing gastric emptying and altering the migration motor complex (MMC), increasing the risk of conditions like gastroparesis and bowel obstruction [28]. Parkman et al.’s [29] recent review noted that liraglutide (long-acting GLP-1 RA) is associated with slower gastric emptying and increasing fasting gastric volume [30]. Overall, 57% of patients treated with liaglutide developed delayed gastric emptying. Kalas et al. similarly reported increased gastroparesis in patients on long-term dulaglutide, liraglutide, and semaglutide therapy [31]. In a multi-database study, Beas et al. found that 41% of patients diagnosed with gastroparesis also had SIBO, highlighting the necessity for further diagnostic exploration [29]. Preclinically, a recent mouse study showed that liraglutide increased the cecal levels of caseinolytic protease B, a component of Escherichia coli, and of norepinephrine. This is the first study to investigate the unique underlying mechanisms related to the effects of GLP-1RA on changes in the gut bacterium [32]. In addition, impaired GI transit favors the proliferation of slow-growing bacterial species in a more stagnant intestinal environment [33]. Other clinical data showed microbiota shifts (fewer beneficial, more harmful bacteria) with constipation and delayed gastric emptying, underscoring the diagnostic significance of microbiota composition changes [34]. Together, these observations suggest that GLP-1-based therapy may predispose to SIBO via motility slowing and harmful bacterial proliferation. Mechanistic studies are still needed to define the specific microbial changes and their clinical implications. Better understanding these mechanisms would greatly enhance diagnostic precision and clinical management strategies.

### 4.5. Study Strength and Limitations

This study has several key strengths. First of all, this study utilizes a large, global real-world dataset (TriNetX; over 163 million patients), supporting broad generalizability and more stable estimates. Secondly, the use of propensity-score matching (1:1) effectively balanced confounders between study cohorts, increasing internal validity of the comparative results. Finally, Real-world databases capture outcomes as they occur in routine practice, offering a view beyond the selective conditions of randomized trials. This data reflects the complexity of everyday care and generally enhances the external validity and generalizability of findings to typical clinical settings.

Despite this strength, our study has several limitations. This retrospective electronic health record study relies on coded diagnoses and test records that vary across sites, so some misclassification and date imprecision are possible. TriNetX lacks structured dietary information; residual confounding by dietary patterns may persist despite propensity matching. TriNetX small-cell suppression (≤10) further limits granularity. Although propensity-score matching balanced measured covariates, unmeasured confounding may remain (e.g., diet, motility disorders, OTC PPIs, probiotics). On the other hand, breath-test substrate/protocols were heterogeneous and not uniformly captured, which may reduce sensitivity; any misclassification is likely non-differential. Finally, the KM module does not provide time-specific numbers at risk; we therefore report start-of-window risk sets. In addition, medication adherence/persistence cannot be fully verified across sites. These limitations temper—but do not negate the consistent direction of the initiator analyses.

### 4.6. Clinical Implicationn and Future Directions

A confirmed SIBO diagnosis is clinically meaningful: it explains bloating, abdominal pain, diarrhea/constipation, malabsorption, weight loss, and micronutrient deficiencies, and it is actionable (dietary measures and targeted antibiotics). Untreated SIBO can lead to repeated visits, labs/imaging, and empiric therapies. Given the low absolute incidence but higher relative hazard among GLP-1/dual users, routine universal screening is not warranted. A targeted, symptom-driven testing strategy (e.g., new/persistent GI symptoms after initiation, suspected delayed gastric emptying, or unexplained nutritional deficiencies) balances diagnostic yield with resource use. Our study did not evaluate cost-effectiveness; however, our recommendations emphasize case-finding rather than universal screening to minimize financial burden.

Moving forward, a prospective study that enrolls new initiators and follows them with standardized breath-testing would provide higher-quality data to confirm these findings. Designs should apply uniform breath-test protocols with centralized adjudication to limit misclassification. Clinically, for patients starting GLP-1 RA or dual GLP-1/GIP RA, symptom-driven SIBO screening by hydrogen breath test may be considered. Additionally, GI motility assessments with microbiome profiling are warranted to investigate the relationship of GLP-1-based therapy, delayed transit, and bacterial overgrowth, which will significantly enhance clinical management and prevention strategies.

In conclusion, to our knowledge, this is the first large-scale study to identify an association between GLP-1 RAs or GLP/GIP RA and an increased incidence of SIBO compared to other second-line diabetes medications. Given the widespread clinical application and proven effectiveness of GLP-1 RAs, GLP/GIP RA for managing T2DM and obesity, clinicians should remain aware of potential gastrointestinal side effects. Early, symptom-driven screening for SIBO with a hydrogen breath test may be warranted in patients initiating these agents. Finally, prospective studies with standardized diagnostic protocols are needed. Future research should focus on mechanisms linking GI motility change, gut microbiota, and SIBO to improve risk stratification and preventive care.

## Figures and Tables

**Figure 1 diagnostics-15-02264-f001:**
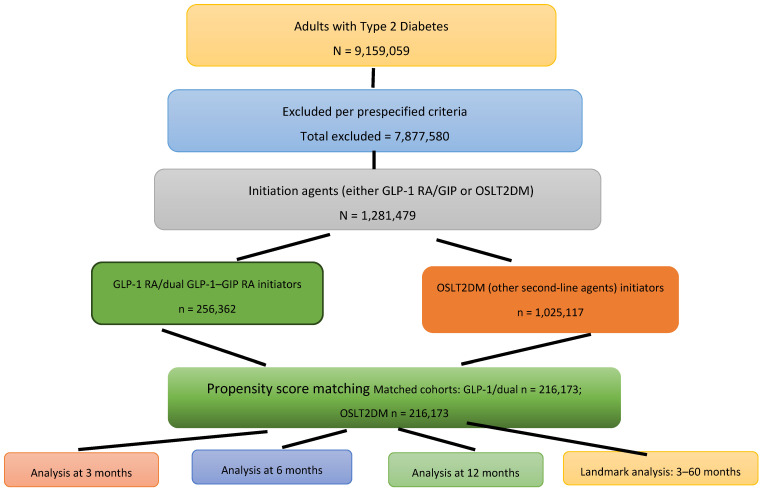
Flow diagram of patient screening, exclusions, and cohort assembly for initiators of GLP-1 receptor agonists or dual GLP-1/GIP receptor agonists (GLP-1/GIP RAs) versus other second-line T2DM agents (OSLT2DM).

**Figure 2 diagnostics-15-02264-f002:**
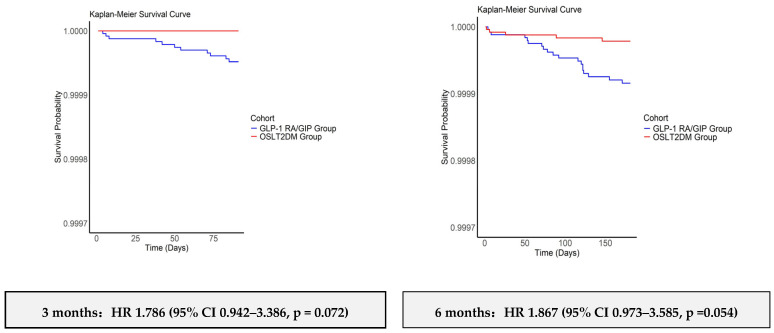
Kaplan–Meier curves for time to small intestinal bacterial overgrowth (SIBO) in propensity-score-matched adults with type 2 diabetes, indicating GLP-1 receptor agonists or dual GLP-1/GIP receptor agonist (GLP-1 RA/GIP) versus other second-line diabetes medications (OSLT2DM). The figures show 3 months, 6 months, 12 months after initiation and 90-day landmark window at 3–60 months. Log rank *p* values and hazard ratios (HRs) with 95% confidence intervals are shown (from univariate Cox proportional hazards models).

**Table 1 diagnostics-15-02264-t001:** Diagnostic, procedure, and medication codes used for query, propensity matching, or outcomes.

Search Terms	Associated Code
Surgery on the Esophagus	CPT: 43279, 43280, 43283, 43284, 43285, 43286, 43287, 43288, 43289, 1007215, 1007219, 1007295 *, 1007298, 1007341, 1020218, 1007335 ICD-10: 0DV1, 0DV2, 0DV3, 0DV4, 0DV5, 0D11, 0D51, 0D52, 0D53, 0D54, 0D55, 0D84, 0DB1, 0DB2, 0DB3, 0DB4, 0DB5, 0DC1, 0DC2, 0DC3, 0DC4, 0DC5, 0DD1, 0DD2, 0DD3, 0DD4, 0DD55, 0DF5, 0DL1, 0DL2, 0DL3, 0DL4, 0DL5, 0DT1, 0DT2, 0DT3, 0DT4, 0DT5, 0D71, 0D72, 0D73, 0D74, 0D75
Surgery on the Stomach	CPT: 1007344, 1007345, 1007352, 1007372, 1007392, 1007385 ICD-10: 0DV6, 0DV7, 0DB6, 0DB7, 0DC6, 0DC7, 0DD6, 0DD7, 0DF6, 0DL6, 0DL7, 0DT6, 0DT7, 0D16
Dyskinesia of Esophagus	IDC-10: K22.4
Scleroderma	IDC-10: M34
Radiation Therapy	TNX Curated: 1001
Lupus Erythematosus	ICD-10: L93
Sjörgen Syndrome	ICD-10: M35.0
Acquired Absence of Other Specified Parts of Digestive Tracts	ICD-10: Z90.49
Other abdominal surgeries	CPT: 1007463, 1007422, 1007570, 1007695, 1007795, 1007908, 1007952, 1007846, 1007854, 1007892, 1018154, 47579, 1007887, 47700, 47701, 47570, 47715, 1007592, 1007596, 1007620, 1007663, 1007671, 1007687, 1007692 SNOMED: 4558008, 80146002, 11466000 ICD-9: 46.2
Ehlers-Danlos Syndrome	ICD-10: Q79.6
Intestinal Adhesions with Obstruction	ICD-10: K56.5
Unspecified Diseases of Spinal Cord	ICD-10: G95
Parkinson’s Disease	ICD-10: G20
Multiple Sclerosis	ICD-10: G35
Amyloidosis	ICD-10: E85
Crohn’s Disease	ICD-10: K50
Ulcerative Colitis	ICD-10: K51
Other Specified Noninfective Gastroenteritis and Colitis	ICD-10: K52.8
Noninfective Gastroenteritis and Colitis, Unspecified	ICD-10: K52.9
Cystic Fibrosis	ICD-10: E84
Other Unspecified Intestinal Obstruction	ICD-10: K56.6
Fistula of Intestine	ICD-10: K63.2
Intestinal Failure	ICD-10: K90.83
Type 2 Diabetes Mellitus	ICD-10: E11
Gastroparesis	ICD-10: K31.84
Glucagon-like Protein-1 Receptor Agonists	ATC: A10BJ RxNorm: 1440051, 1551291, 1991302, 2601723, 475968, 60548 **
Sulfonylureas	ATC: A10BB
Dipeptidyl Peptidase 4 Inhibitors	ATC: A10BH
Sodium-Glucose Co-Transporter 2 Inhibitors	ATC: A10BK
Thiazolidinediones	ATC: A10BG

* Denotes codes that include all subcodes except codes that are denoted by’. ** Denotes a code that had a TriNetX filter attached to it to differentiate between its two formulations. ICD-10 = International classification of diseases, version 10, CPT = current procedural terminology, ATC = anatomical therapeutic chemical classification system, TNX = TriNetX curated codes, RxNorm = US normalized drug names.

**Table 2 diagnostics-15-02264-t002:** Baseline characteristics of patients initiating GLP-1 receptor agonists or dual GLP-1/GIP receptor agonists (GLP-1/GIP RAs) versus other second-line type 2 diabetes medications (OSLT2DM), before and after propensity-score matching. Note: Per TriNetX privacy policy, *: cells with ≤10 patients are masked (displayed as “≤10” or 10); exact counts below 10 are not disclosed.

	Before Matching			After Matching		
	Cohort, No. (%)			Cohort, No. (%)		
Demographics	GLP-1 RA	Control	Std Diff.	GLP-1 RA	Control	Std Diff.
Total No.	234,701	952,418		216,173	216,173	
Age, mean (SD), y	55.6 ± 13.5	62.5 ± 12.6	0.5279	56.7 ± 13.0	56.7 ± 13.4	0.002
Sex						
Female	130,070 (55.4%)	394,335 (41.4%)	0.2833	115,650 (53.5%)	115,379 (53.4%)	0.0025
Male	94,047 (40.1%)	535,217 (56.2%)	0.327	91,358 (42.3%)	91,454 (42.3%)	0.0009
Ethnicity						
Hispanic/LatinX	19,273 (8.2%)	86,609 (9.1%)	0.0314	18,213 (8.4%)	17,304 (8.0%)	0.0153
Not Hispanic/LatinX	137,998 (58.8%)	485,443 (51.0%)	0.1578	125,595 (58.1%)	131,526 (60.8%)	0.0559
Unknown	77,430 (33.0%)	380,366 (39.9%)	0.1447	72,365 (33.5%)	67,343 (31.2%)	0.0497
Race						
Asian	8366 (3.6%)	79,874 (8.4%)	0.2045	8231 (3.8%)	7947 (3.7%)	0.0069
Black	44,108 (18.8%)	132,399 (13.9%)	0.1326	39,336 (18.2%)	39,746 (18.4%)	0.0049
White	129,966 (55.4%)	459,037 (48.2%)	0.144	119,423 (55.2%)	123,649 (57.2%)	0.0394
American Indian or Alaska Native	792 (0.3%)	2606 (0.3%)	0.0116	741 (0.3%)	577 (0.3%)	0.0138
Native Hawaiian or Other Pacific Islander	1430 (0.6%)	5888 (0.6%)	0.0011	1332 (0.6%)	1566 (0.7%)	0.0133
Unknown	40,556 (17.3%)	234,549 (24.6%)	0.1813	38,337 (17.7%)	34,059 (15.8%)	0.053
Diabetic Comorbidities						
Primary hypertension	159,587 (68.0%)	640,293 (67.2%)	0.0164	148,292 (68.6%)	148,742 (68.8%)	0.0045
Hyperlipidemia	109,315 (46.6%)	401,625 (42.2%)	0.0888	100,947 (46.7%)	101,353 (46.9%)	0.0038
Atherosclerotic heart disease of native coronary artery	31,441 (13.4%)	205,487 (21.6%)	0.2166	30,824 (14.3%)	30,893 (14.3%)	0.0009
Other hypothyroidism	35,475 (15.1%)	91,180 (9.6%)	0.1691	30,643 (14.2%)	30,511 (14.1%)	0.0018
Type 2 diabetes mellitus with neurological complications	28,515 (12.2%)	95,272 (10.0%	0.0684	26,598 (12.3%)	26,714 (12.4%)	0.0016
Chronic kidney disease	26,023 (11.1%)	147,717 (15.5%)	0.1305	25,181 (11.6%)	25,195 (11.7%)	0.0002
Heart failure	16,791 (7.2%)	129,371 (13.6%)	0.2121	16,593 (7.7%)	17,064 (7.9%)	0.0081
Other functional intestinal disorders	18,922 (8.1%)	57,691 (6.1%)	0.0783	16,232 (7.5%)	15,598 (7.2%)	0.0112
Peripheral vascular disease, unspecified	8504 (3.6%)	42,087 (4.4%)	0.0405	8128 (3.8%)	7986 (3.7%)	0.0035
Atherosclerosis	8079 (3.4%)	40,400 (4.2%)	0.0416	7757 (3.6%)	7631 (3.5%)	0.0031
Cerebral infarction	7801 (3.3%)	50,448 (5.3%)	0.0973	7609 (3.5%)	7467 (3.5%)	0.0036
End-stage renal disease	3604 (1.5%)	17,977 (1.9%)	0.0271	3414 (1.6%)	3417 (1.6%)	0.0001
Fibrosis and cirrhosis of liver	2979 (1.3%)	13,675 (1.4%)	0.0144	2824 (1.3%)	2667 (1.2%)	0.0065
Other immunodeficiencies	2369 (1.0%)	4166 (0.4%)	0.0675	1839 (0.9%)	1984 (0.9%)	0.0072
Malnutrition	1716 (0.7%)	14,370 (1.5%)	0.0739	1671 (0.8%)	1505 (0.7%)	0.009
Opioid dependence	1762 (0.8%)	4262 (0.4%)	0.0393	1515 (0.7%)	1500 (0.7%)	0.0008
Human immunodeficiency virus (HIV) disease	1376 (0.6%)	3418 (0.4%)	0.0332	1189 (0.6%)	1271 (0.6%)	0.005
Drug-induced constipation	1078 (0.5%)	2148 (0.2%)	0.04	894 (0.4%)	875 (0.4%)	0.0014
Celiac disease	614 (0.3%)	658 (0.1%)	0.0474	442 (0.2%)	289 (0.1%)	0.0172
Other chronic pancreatitis	408 (0.2%)	3720 (0.4%)	0.0409	404 (0.2%)	467 (0.2%)	0.0065
Immunodeficiency with predominantly antibody defects	361 (0.2%)	812 (0.1%)	0.0198	305 (0.1%)	249 (0.1%)	0.0072
Diverticular disease of small intestine without perforation or abscess	183 (0.1%)	776 (0.1%)	0.0012	172 (0.1%)	154 (0.1%)	0.003
Common variable immunodeficiency	100 (>0.1%)	181 (>0.1%)	0.0135	83 (>0.1%)	65 (>0.1%)	0.0045
Acromegaly and pituitary gigantism	92 (>0.1%)	194 (>0.1%)	0.0109	76 (>0.1%)	67 (>0.1%)	0.0023
Alcohol-induced chronic pancreatitis	42 (>0.1%)	155 (>0.1%)	0.0207	42 (>0.1%)	92 (>0.1%)	0.0131
Achlorhydria	39 (>0.1%)	47 (>0.1%)	0.0113	36 (>0.1%)	17 (>0.1%)	0.0079
Myotonic muscular dystrophy	25 (>0.1%)	111 (>0.1%)	0.0009	23 (>0.1%)	28 (>0.1%)	0.0021
Combined immunodeficiencies	21 (>0.1%)	58 (>0.1%)	0.0033	16 (>0.1%)	15 (>0.1%)	0.0005
Tropical sprue	10 * (>0.1%)	12 (>0.1%)	0.0057	10 (>0.1%)	10 * (>0.1%)	0.0001
Gastroenteritis and colitis due to radiation	11 (>0.1%)	46 (>0.1%)	0.0002	10 * (>0.1%)	12 (>0.1%)	0.0013
Systemic sclerosis (scleroderma)	0 (0.0%)	0 (0.0%)	0	0 (0.0%)	0 (0.0%)	0
Amyloidosis	0 (0.0%)	0 (0.0%)	0	0 (0.0%)	0 (0.0%)	0
Gastroparesis	0 (0.0%)	0 (0.0%)	0	0 (0.0%)	0 (0.0%)	0
Labs						
BMI	37.2 ± 8.33	31.5 ± 7.52	0.7151	36.4 ± 7.9	35.9 ± 8.31	0.0595
At least 25 kg/m^2^	168,304 (71.7%)	492,218 (51.7%)	0.421	150,141 (69.5%)	150,743 (69.7%)	0.0061
At least 30 kg/m^2^	150,585 (64.2%)	345,445 (36.3%)	0.5809	132,446 (61.3%)	132,557 (61.3)	0.0011
At least 35 kg/m^2^	111,793 (47.6%)	188,694 (19.8%)	0.6157	94,275 (43.6%)	94,681 (43.8%)	0.0038
At least 40 kg/m^2^	70,047 (29.8%)	95,351 (10.0%)	0.5126	55,850 (25.8%)	56,129 (26.0%)	0.0029
At least 45 kg/m^2^	38,925 (16.6%)	44,772 (4.7%)	0.3927	29,328 (13.6%)	29,483 (13.6%)	0.0021
At most 50 kg/m^2^	164,607 (70.1%)	554,881 (58.3%)	0.2496	149,230 (69.0%)	148,578 (68.7%)	0.0065
At most 45 kg/m^2^	155,179 (66.1%)	542,021 (56.9%)	0.1901	141,532 (65.5%)	140,779 (65.1%)	0.0073
At most 40 kg/m^2^	135,445 (57.7%)	508,701 (53.4%)	0.0866	124,545 (57.6%)	123,635 (57.2%)	0.0085
At most 35 kg/m^2^	100,515 (42.8%)	439,347 (46.1%)	0.0665	93,690 (43.3%)	92,811 (42.9%)	0.0082
At most 30 kg/m^2^	52,960 (22.6%)	298,935 (31.4%)	0.1998	50,079 (23.2%)	49,462 (22.9%)	0.0068
Hemoglobin—A1C	7.79 ± 2.16	8.1 ± 2.18	0.1445	7.88 ± 2.17	7.95 ± 2.25	0.0313
At least 7.0%	105,322 (44.9%)	357,521 (37.5%)	0.1495	98,200 (45.4%)	96,673 (44.7%)	0.0142
At least 7.5%	87,847 (37.4%)	298,785 (31.4%)	0.1278	82,081 (40.0%)	81,397 (37.7%)	0.0065
At least 8.0%	75,221 (32.1%)	248,228 (26.1%)	0.1321	70,059 (32.4%)	69,792 (32.3%)	0.0026
At least 8.5%	64,362 (27.4%)	203,207 (21.3%)	0.1421	59,668 (27.6%)	59,585 (27.6%)	0.0009
At least 9.0%	56,002 (23.9%)	170,295 (17.9%)	0.1476	51,692 (23.9%)	51,718 (23.9%)	0.0003
At least 9.5%	47,943 (20.4%)	141,157 (14.8%)	0.1475	44,070 (20.4%)	44,053 (20.4%)	0.0002
At least 10.0%	41,338 (17.6%)	118,895 (12.5%)	0.1438	37,856 (17.5%)	37,817 (17.5%)	0.0005
At most 11.0%	150,745 (64.2%)	432,191 (45.4%)	0.3857	133,601 (61.8%)	129,609 (60.0%)	0.0378
At most 10.5%	148,443 (63.2%)	422,823 (44.4%)	0.3851	131,369 (60.8%)	127,446 (59.0%)	0.037
At most 10.0%	145,620 (62.0%)	411,384 (43.2%)	0.3845	128,625 (59.5%)	124,702 (57.7%)	0.0368
At most 9.5%	142,545 (60.7%)	399,096 (41.9%)	0.3836	125,658 (58.1%)	121,680 (56.3%)	0.0372
At most 9.0%	138,532 (59.0%)	383,069 (40.2%)	0.3829	121,816 (56.4%)	117,808 (54.5%)	0.0373
At most 8.5%	133,853 (57.0%)	363,584 (38.2%)	0.3845	117,348 (54.3%)	113,434 (52.5%)	0.0363
At most 8.0%	129,033 (55.0%)	343,442 (36.1%)	0.3869	112,735 (52.2%)	108,934 (50.4%)	0.0352
Medications						
Opiod analgesics	108,932 (46.4%)	381,176 (40.0%)	0.1293	98,570 (45.6%)	99,173 (45.9%)	0.0056
Proton pump inhibitors	71,525 (30.5%)	267,157 (28.1%)	0.0533	64,820 (30.0%)	65,383 (30.2%)	0.0057
Dicyclomine	4813 (2.1%)	9964 (1.0%)	0.0814	3952 (1.8%)	3864 (1.8%)	0.0031
Loperamide	3068 (1.3%)	11,347 (1.2%)	0.0104	2749 (1.3%)	2600 (1.2%)	0.0062
Diphenoxylate	1123 (0.5%)	2909 (0.3%)	0.0277	945 (0.4%)	940 (0.4%)	0.0004

## Data Availability

All raw output data from TriNetX are available on reasonable requests by contacting the corresponding author.

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
