# Peer review of "Diagnostic Evaluation of an Increased Risk of Developing Small Intestinal Bacterial Overgrowth Associated with Glucagon-like Peptide-1 (GLP-1) Receptor Agonists and Dual GLP-1/GIP Receptor Agonists: A Global Retrospective Multicenter Cohort Analysis"

_diagnostics, 2025, doi:10.3390/diagnostics15172264_

Round 1

Reviewer 1 Report

Comments and Suggestions for Authors

1. Please clarify the methodology for calculating the event incidence rate (lines 212–214) and specify the statistical techniques employed to compare intergroup differences.
2. No results from subgroup analyses were reported.
3. The long-term and short-term outcomes based on Kaplan-Meier (KM) estimates are not adequately illustrated. It is recommended to apply landmark analysis for improved interpretation.
4. Cox proportional hazards modeling with adjustment for confounding factors was not implemented.
5. Kindly provide the results of the Cox proportional hazards (CoxPH) test.
6. On line 220, please report the median follow-up duration instead of the mean follow-up time.
7. Please include t-tests or chi-square tests for correlation in the baseline data before and after matching. Additionally, carefully evaluate the presentation of baseline characteristics and intergroup comparisons—for instance, whether the data conform to a normal distribution and whether using standard deviation (std) alone is sufficient for comparing group differences.
8. Why was a caliper width of 0.1 selected for propensity score matching (PSM)? Is there supporting literature for this choice? Were alternative weighting methods, such as inverse probability weighting (IPTW) or overlapping weight adjustment, considered?
9. Please provide a completed checklist of the STROBE (Strengthening the Reporting of Observational Studies in Epidemiology) guidelines.
10. The current flowchart is visually unprofessional and should be redrawn with straight and properly aligned lines.
11. Please include the number of subjects at risk at each time point on the Kaplan-Meier survival curves.

Reviewer 2 Report

Comments and Suggestions for Authors

Current report investigated the risk of Glucagon-like peptide-1 receptor agonists (GLP-1 RAs) on small intestinal bacterial overgrowth (SIBO) using the TriNetX global database. Please conduct the concerns below.

  1. In the introduction section, it seems better to introduce the details of reference 13 in addition to reference 14 for association between GLP-1 RAs and SIBO.
  2. Short-term cohorts must follow the previous report with reference.
  3. Definition of “Other Second-line Diabetes Medications (OSLT2DM)” in Figure 2 needs to indicate in clear.
  4. Clinicians are encouraged to proactively implement diagnostic screening strategies for SIBO, particularly using breath tests, for patients initiating GLP-1 RA therapy. It seems the merit of current report with details. Please improve this statement.
  5. Reliability of this analysis was ignored. Why?

Reviewer 3 Report

Comments and Suggestions for Authors

The growing prevalence of obesity and type 2 diabetes mellitus (T2DM) represents a significant challenge to the healthcare system, and their effective management is critical for reducing economic burdens. Currently, Glucagon-Like Peptide-1 (GLP-1) Receptor Agonists and Dual GLP-1/GIP Receptor A agonists have garnered considerable attention as a promising therapeutic option for both obesity and T2DM. However, with the escalating use of these medications, partly driven by widespread advertising, a comprehensive focus on both their benefits and potential side effects has become imperative. The study conducted by Sun et al. aimed to investigate the impact of GLP-1 RA/GIP analogs on the risk of SIBO in T2DM patients. The findings indicated a statistically significant increase in SIBO risk associated with GLP-1 analog use in the short term, while this association was found to be non-significant over the long term. Nevertheless, several concerns regarding the study remain.

- Although the writing is generally understandable, the English language quality of the manuscript is rather poor, and so a professional editing of the manuscript to bring the English up to the correct standard is recommended.

- Do breath tests possess sufficient sensitivity for SIBO diagnosis? What specific type of breath test was utilized? Further details regarding the application of this diagnostic test are required.

- The precise age range of the participating patients needs to be specified. To what exact number does the term 'older' refer?

- The inclusion and exclusion criteria require greater detail, particularly concerning the control group.

- Are patients evaluated for medullary thyroid carcinoma? Was the intake of probiotics or prebiotics assessed?

- Do the drugs in the OSLT2DM group all have a uniform effect on the prevalence of SIBO? It is essential that the results for each different medication be presented.

- Were patients evaluated for their dietary intake? This is crucial since various diets have different effects on SIBO.

- The table captions must be revised, as they are highly ambiguous and fail to provide essential information. Furthermore, the captions should be moved below the tables, and all abbreviations must be defined.

- Table 2 should be moved to the findings section. It is very long and ambiguous.

- What is the clinical significance of a SIBO diagnosis? Given that there were only 36 and 82 diagnosed cases in the GLP and control groups, respectively, does the application of this diagnostic approach not impose a significant financial burden on the healthcare system?

- Moreover, the low prevalence of SIBO may compromise the reliability and generalizability of the findings.

- This study has further limitations that must be thoroughly discussed.

- The discussion section is notably weak and not structured within a proper framework. It must be substantiated with robust evidence from the findings. It MUST be revised comprehensively.

Round 2

Reviewer 3 Report

Comments and Suggestions for Authors

It was revised successfully.

Please clarify the phrase "deceased patients at baseline" in the method section. It is too vague. 

Author Response

Comments: Please clarify the phrase "deceased patients at baseline" in the method section. It is too vague. 

Thanks for the comments. “deceased patients at baseline” we mean patients recorded as dead on or before the cohort entry (“index”) date. Because such patients are not at risk for a new outcome after index, we excluded them to ensure valid time-to-event analyses.